# Laparoscopic Management of Multiple Liver, Omental, Mesenteric, Peritoneal, and Round Ligament Hydatid Cysts—A Rare Report of a Case and a Systematic Literature Review

**DOI:** 10.3390/jpm14020205

**Published:** 2024-02-14

**Authors:** Alin Mihetiu, Dan Georgian Bratu, Ciprian Tanasescu, Bogdan Ioan Vintilă, Alexandra Sandu, Mariana Sandu, Dragos Serban, Dan Sabau, Adrian Hasegan

**Affiliations:** 1County Clinical Emergency Hospital of Sibiu, 550245 Sibiu, Romania; alin.mihetiu@ulbsibiu.ro (A.M.); ciprian.tanasescu@ulbsibiu.ro (C.T.); bogdan.vintila@ulbsibiu.ro (B.I.V.); alexandrae.sandu@ulbsibiu.ro (A.S.); dan.sabau@ulbsibiu.ro (D.S.); adrian.hasegan@ulbsibiu.ro (A.H.); 2Faculty of Medicine, Lucian Blaga University of Sibiu, 550169 Sibiu, Romania; 3Faculty of Medicine, Carol Davila University of Medicine and Pharmacy Bucharest, 020021 Bucharest, Romania; dragos.serban@umfcd.ro

**Keywords:** hydatid cyst, laparoscopy, intra-abdominal hydatidosis

## Abstract

Hydatid cyst disease is a parasitic ailment with an endemic nature, predominantly affecting geographical areas with a tradition in animal husbandry. The most common localization of hydatid disease is in the liver (60%), followed by the lungs, with other organ localizations comprising less than 10%. The surgical approach to this condition can be carried out through open surgery or laparoscopy. The coexistence of hepatic and intraperitoneal hydatidosis often leads to the preference for open surgery. We performed a literature review aiming to retrieve data regarding demographic characteristics, clinical features, preoperative management, and surgical approach concerning these unusual localizations of hydatid disease. It was observed that the mesenteric localization frequently presented with acute abdominal pain (*p* = 0.038) and that the open approach was preferred in 85.71% of cases. Furthermore, an interdependence was identified between the localization of the cysts and the type of surgical approach (*p* = 0.001), with mesenteric localizations being approached through laparotomy and excision (*p* = 0.037), while omental localizations, due to the easier approach, benefited from laparoscopy with excision in 14.29% of cases. Overall, the laparoscopic approach was less frequently used, but its utilization resulted in a lower number of complications and faster recovery. Additionally, we present a rare case of hepatic and intra-abdominal hydatidosis, resolved exclusively through a laparoscopic approach, including a review of the literature for these uncommon localizations of hydatid disease. A 45-year-old patient diagnosed with multiple hydatid cysts, both hepatic and intraperitoneal, underwent surgical intervention with exploratory laparoscopy. Laparoscopic excision of peritoneal, epiploic, mesenteric cysts, and round ligament, along with laparoscopic inactivation, evacuation, and pericystectomy of hepatic hydatid cysts, was performed. The patient’s recovery was uneventful, and she was reevaluated at 3 and 9 months without signs of recurrence. The association of hepatic hydatid cysts with multiple intra-abdominal localizations is not commonly encountered. The treatment of choice is surgical and is predominantly conducted through open surgery. The presented case is unique due to the exclusive laparoscopic approach in the management of mixed hepatic and intra-abdominal hydatidosis.

## 1. Introduction

*Echinococcus granulosus* is a tapeworm that, in its larval stage, causes hydatid disease. Hydatidosis is an endemic condition prevalent in geographic areas such as the Middle East, Asia, sub-Saharan Africa, the Mediterranean region, South America, and the Balkans. The tapeworm localized itself in the intestinal tract of canids or wild carnivores, while sheep and humans act as intermediate hosts through accidental ingestion of the tapeworm eggs. *Echinococcus granulosus* and *Echinococcus multilocularis* are the most common subtypes encountered in medical practice. Following the ingestion, the tapeworm eggs penetrate the intestinal lamina propria, entering the vascular or lymphatic system and disseminating to various organs, where they develop into hydatid cysts. The most frequent localization of the hydatid cyst is in the liver (70%), followed by the lungs (20%), with other locations (brain, eyes, bones, muscles, skin, etc.) totaling 10% [1].

Due to considerations related to the portal laminar flow, the larger size of the right portal pedicle, and the portal capillary filter, the right hepatic lobe is more frequently affected by hydatid disease (65%) compared to the left lobe (25%). Transition zones between the two lobes and bilateral placement are found in a10% proportion. In 85% of cases, hepatic hydatid cysts are singular, while in 15% of cases, they are multiple. Between 20 and 40% of patients may develop complications of hydatid disease [1].

The highly immunogenic nature of the cyst content can lead to anaphylactic reactions, including anaphylactic shock, more commonly observed in cases of intraperitoneal or intravascular rupture of the cyst. Intraperitoneal rupture can complicate hydatid peritonitis or peritoneal hydatidosis. A particular situation is represented by the intact migration of the cyst from the hepatic to the peritoneal level, a situation known as an aborted cyst [1,2].

The rupture of the hydatid cyst into the bile ducts is a common complication of hydatid disease and leads to mechanical jaundice, acute pancreatitis, cholangitis, or, in prolonged courses, biliary cirrhosis. Other complications are compressive ones with vascular effects (ischemic or portal hypertension-related, Budd–Chiari syndrome), effects on the biliary tract (jaundice), or neighboring organs [2,3].

Intra-abdominal, extrahepatic locations of the hydatid cyst are rare. The spleen is, after the liver, the intra-abdominal organ with the most frequent localization of the hydatid cyst (4% of the total hydatid cyst locations in humans) [4]. Peritoneal hydatid cyst is found in 5–16% of abdominal locations and is often secondary to dissemination originating from the liver. Primary cysts are rare and occur in 2% of cases. Mesenteric and round ligament locations are extremely rare [5,6,7].

Complications in these locations are similar to hepatic localizations and can include rupture into the abdominal cavity, organ compressions, and vascular compressions. Spontaneous fistulization into a body cavity, although rarely described, is still possible. Despite a decreasing incidence of this condition, the frequency of cases in endemic areas makes hydatidosis a public health issue. Symptoms are nonspecific, with abdominal pain being the most commonly implicated, while complicated cases may present with signs of acute abdomen, shock, or jaundice.

Imaging diagnosis is sometimes incidental through ultrasound or CT examinations, with cystic formations exhibiting particular characteristics that classify them according to the Gharbi or WHO classification [8].

Pharmacological therapy currently involves medications from the benzimidazole group, predominantly albendazole (ABZ) or its combination with praziquantel. For disseminated hydatidosis, nitazoxanide may be included in the treatment plan [9].

Alone, pharmacological treatment can reduce the viability of hydatid cysts but is dependent on the size and number of the cysts. There are not enough studies to demonstrate inactivation, regression, and absence of recurrence on a large scale in pharmacological management without the association of an invasive procedure. For such therapy to be effective, it requires prolonged treatment, high costs, and management of adverse reactions, especially hepatotoxicity. Combining albendazole as adjuvant or neoadjuvant therapy with Puncture, Aspiration, Instillation, Reaspiration (PAIR), Modified Catheterisation Technique (MoCAT), or surgery represents the most optimal option in terms of cyst inactivation and low recurrence [10,11,12,13].

When comparing albendazole with mebendazole, albendazole shows better results, and its combination with praziquantel is the most efficient drug therapy [9].

Introduced in the early 1990s into the therapeutic management of hepatic hydatid cysts, PAIR is currently the main invasive method for approaching the cyst. The method has undeniable benefits compared to the surgical approach but has limitations regarding the localization of hepatic hydatid cysts, their size, the possibility of optimal pericystic isolation, and recurrence [14,15,16].

To achieve a unified management of this condition, an algorithm has been developed that establishes a therapeutic sequence based on the imaging classification of the hydatid cyst and its dimensions (Figure 1).

The previously mentioned algorithm is more applicable to hepatic, pulmonary, or splenic localization of echinococcosis, but disseminated intra-abdominal hydatidosis rarely benefits from PAIR and extremely rarely benefits from laparoscopic surgery.

Peritoneal, mesenteric, and omental localizations, especially when multiple or associated with hepatic hydatid disease, typically are an indication for open surgery. This may represent the sole method of treatment in the majority of cases, being more convenient for the surgeon, with lower risks of intra-abdominal contamination or missing a cystic formation. However, it comes with the disadvantages of open surgery in terms of recovery and immediate and long-term postoperative complications.

The therapeutic management of this condition is complex and involves not only antiparasitic treatment but also endoscopic, interventional radiology, or surgical approaches.

The present study aims to analyze the data from the medical literature with similarities regarding this particular type of abdominal hydatidosis. Considering that frequently, multiple extrahepatic localizations require open surgery, we believe that attempting a minimally invasive approach is necessary, as it has undeniable benefits for the patient.

The particularity of this case lies in the laparoscopic approach, which addressed both hepatic hydatidosis and other abdominal localizations. In the existing literature, there are limited instances where surgical intervention for a pathology of such an extensive nature, accompanied by numerous secondary dissemination, has been successfully conducted using minimally invasive methods.

## 2. Materials and Methods

We present the case of a patient with multiple hepatic and intraperitoneal localizations of hydatid disease, whose surgical management was performed laparoscopically.

As these localizations are relatively rare in the course of hydatid disease, we conducted a literature review to observe the relationship between localization, symptoms, type of scolicidal agent, and surgical treatment used.

By accessing the PubMed and Google Scholar databases and entering the terms “hydatid cyst” and “mesentery”, “hydatid cyst” and “omentum”, and “hydatid cyst” and “round ligament”, we obtained 161 results in PubMed and 431 results in Google Scholar.

Although a specific time interval was not set, all results were obtained for the last 20 years. After selecting the results, 14 articles corresponding to PubMed and 7 to Google Scholar were identified.

To reduce the risk of potential bias, two authors independently evaluated the results. The agreement ratio for included articles was 98%, and for excluded articles, it was 95%.

However, even under these conditions, data interpretation can be subject to bias. Therefore, a few comments on certain aspects of the study are necessary.

Cases presenting strictly peritoneal localization or the association of hepatic hydatidosis with peritoneal hydatidosis were not included in this study because such situations are not rare, with peritoneal implantation being the most common area of dissemination for intraabdominal hydatid rupture.

Introducing data on strictly peritoneal localization or the association of hepatic hydatidosis with peritoneal hydatidosis into this study would not have contributed to achieving this study’s objective, which is to emphasize the importance of a laparoscopic approach, even in novel or challenging disease localizations.

In some of the articles included in this study, preoperative antiparasitic drug therapy is not mentioned, and in some cases, the type of hospital admission presentation (emergency or elective) is not specified. Based on the description of symptoms and laboratory data, as well as overlap with cases where preoperative therapy is not mentioned, it was concluded that these cases presented as emergencies. In such situations, antiparasitic therapy can be initiated postoperatively after resolving the surgical emergency.

Additionally, not all results specify the type of scolicidal agent used to inactivate the parasite. In this situation, unable to identify or extrapolate the type of scolicide, the results were processed strictly based on articles that specified the nature of the agent.

The rarity of cases with such localizations did not allow consulting extensive studies or systematic reviews on the subject, resorting instead to case report articles, the number of which did not permit exhaustive statistics. This article adhered to the PRISMA guidelines for reporting systematic reviews, and the PRISMA checklist was completed for the manuscript and abstract. We gathered data and employed SPSS Data Analysis Software 28.0.1 and Datatab Team (2024).DATAtab: Online Statistics Calculator. DATAtab e.U., Graz, Austria. URL https://datatab.net) accessed on 18 December 2023 for the analysis and interpretation of information obtained from abovementioned sources, leading to evidence-based conclusions. Because our review was conducted retrospectively, we were unable to register our study in the PROSPERO database, as it is specifically designed for prospective studies.

The data retrieved from the study were synthesized using the PRISMA flow chart (Figure 2) [18].

The 21 articles resulting from the selection (14 from PubMed and 7 from Google Scholar) were analyzed by evaluating data related to the symptoms present at admission, diagnostic methods, preoperative pharmacological therapy, localization of hydatid cysts, scolicidal agent used for inactivation, and the type of surgical intervention (Table 1).

## 3. Results

The average age was 42.33 ± 18.78, 42.54 ± 20.1 for males and 42 ± 17.74 for females; 61.9% were males (*n* = 13), and 38.1% were females (*n* = 8). The most frequent symptom was abdominal pain (47.62%), with 9.52% of cases having an acute character.

A total of 28.57% of cases presented with the sole symptom of abdominal mass, and 14.3% (*n* = 3) had associated pain and the presence of an abdominal tumor.

Ultrasound in association with CT scan (33.3%) was the main diagnostic imaging sequence, followed by ultrasound and CT scan (28.6%), with one case being diagnosed with CT scan and later MRI, and in one case, the imaging method was not mentioned.

In 33.33% (*n* = 7) of the patients included in this study, preoperative therapy with albendazole was initiated. Other types of preoperative drug therapy (praziquantel, mebendazole, or combinations) were not identified.

The low reporting of preoperative pharmacological therapy can be explained by the symptomatology characterized by abdominal pain, sometimes acute, with a preference for initiating surgical treatment as early as possible to avoid cyst complications.

Preoperative antiparasitic treatment, given the specific symptomatology, is debatable due to the prolonged duration required for the treatment to be effective. In all postoperative cases, pharmacological therapy was administered.

The most common localization of hydatid cyst was in the mesentery (47.62%), followed by the omentum (23.81%), with multiple localizations observed in 23.8% of patients.

The rarest location was in the round ligament, identified in a single case.

Analyzing the relationship between localization and symptomatology, the presence of acute pain was observed for mesenteric localizations (OR = 2.2, z statistic = 2.08, *p* = 0.038, 95% CI = 0.04–0.91), with abdominal pain also being the most common symptom in mesenteric localizations (28.57%). The association of pain with palpable tumor formation in hydatid cysts located in the mesentery was observed in 4.76% of cases (OR = 0.06, z statistic = 2.69, *p* = 0.007, 95% CI = 0.0–0.47). The variables with statistical significance were marked in tables with *.

Multiple localizations were observed in this study (19.04%), with the most frequently involved parenchymal organ being the liver (n = 2), followed by the spleen. In these localizations, the presenting symptom was abdominal pain, which was related to the secondary localization (omentum and mesentery) (Figure 3).

Regarding the solution for inactivating the parasite, in 66.67% (n = 14) of the analyzed cases, the type of solution was not specified; in 19.05% (n = 4), 10% povidone-iodine was used; and in 14.28% (n = 3), hypertonic saline was employed (Table 2).

Analyzing the types of surgical interventions, a higher proportion of open surgery interventions (85.71%) was observed compared to laparoscopic approaches (14.29%).

The most frequently used technique was laparotomy with hydatid cyst excision (52.38%), followed by laparotomy and drainage (9.52%) and laparotomy with partial cyst excision (9.52%).

Regarding the attitude towards the cyst, complete excision was preferred in 85.71% of cases, while partial excision or pericystectomy was chosen for the remaining cases.

Cases with associated hepatic or splenic localization were treated either by partial excision, cyst drainage, or even hepatectomy in open surgery. In a single case, laparoscopic excision of the hepatic cyst, as well as peritoneal and greater omentum localizations, was performed (Table 3).

Comparing the type of surgical intervention and the topography of the hydatid cyst, a significant interdependence was observed between the surgical intervention and the localization of hydatid cysts (*p* = 0.001) (Figure 4). There was a correlation between mesenteric localization and laparotomy with drainage (OR 0.11, z statistic =2.08, *p* = 0.037, 95% CI 0.01–0.88) and laparotomy with partial excision (OR 0.11, z statistic= 2.08, *p* = 0.037, 95% CI 0.01–0.88).

Mesenteric localization predominantly benefited from laparotomy and excision (38.1%), while laparoscopy with complete excision was used for omental localizations (14.29%).

The argument for this approach lies in the easier access to the greater omentum compared to the mesentery. Laparoscopy was used for hydatid cyst of the greater omentum in 9.52% of cases, addressing multiple localizations, omental, peritoneal, and hepatic, in 4.76% of cases.

## 4. Discussion

Peritoneal localizations of echinococcosis are more common than omental or mesenteric ones. The most frequent mechanism of occurrence is through intraperitoneal seeding following the rupture of a hepatic hydatid cyst. Primary peritoneal hydatidosis is rare. These types of abdominal localizations are extremely rare, with only 49 such cases described in the specialized literature, involving peritoneal, epiploic, or mesenteric locations. The localization in the round ligament corresponds to only one reported case in the literature [20,28].

Our study also revealed only two cases where the laparoscopic approach was used for intraperitoneal localizations: one case with a single omental localization and another with a single hepatic and omental localization and multiple peritoneal localizations [36,37].

None of the obtained reports indicated the association of multiple hepatic hydatidoss with the localizations mentioned in the case reports that benefited from laparoscopic treatment.

One of the challenging intraoperative complications in hydatid cyst surgery is the contamination of the surgical field.

Scolicidal solutions play a role in inactivating the parasitic organism within the cyst, preventing potential contamination in the abdominal cavity. Isolating the surgical field with cloths soaked in scolicidal solutions provides an additional mechanism for intraoperative protection, a measure not applicable in the PAIR approach.

The chemical agents used must have good penetrability and effectiveness in inactivating the parasite without causing local, organ, or systemic adverse reactions. Scolicidal substances include formaldehyde, absolute alcohol, hypertonic saline, silver nitrate, chlorhexidine, cetrimide, hydrogen peroxide, and povidone-iodine.

While formaldehyde and silver nitrate were widely used in the past, they have gradually been abandoned, not due to their low antiparasitic efficacy but because of the frequent complications associated with their use.

Complications can range from cholangitis (formaldehyde has a definite cytolytic effect on biliary epithelium) to methemoglobinemia (with the use of silver nitrate), biliary cirrhosis, liver failure, or even death [38,39,40].

The use of hydrogen peroxide was quickly abandoned due to the increased risk of embolism. Cetrimide is mainly used in combination with chlorhexidine and has proven to be effective in inactivation. Chlorhexidine, when used in medical practice, typically has a concentration ranging from 0.04% to 0.08%, a concentration with the best efficiency-to-toxicity ratio. However, both substances carry a risk of cholangitis [12,41].

Povidone-iodine has good efficacy as a scolicide, but it carries an increased risk of injury to the bile duct, hepatotoxicity, and nephrotoxicity [40].

The WHO Informal Working Group on Echinococcosis (WHOIWGE) recommends hypertonic saline 20% as the optimal protoscolicidal agent in surgery and either hypertonic saline 20% or ethyl alcohol 95% in PAIR, as these substances are effective with fewer complications [5,42,43].

Surgical intervention for hepatic hydatid cysts is recommended for cysts larger than 5 cm and in locations difficult to approach with PAIR.

Although PAIR is the initial invasive approach for hydatid cysts, offering indisputable benefits such as reduced hospitalization, low postoperative pain, and reduced postoperative complications, laparoscopy remains an alternative with a lower recurrence rate [14,44].

The ability to create a protective field (soaked in scolicidal agent) around the cyst during the surgical approach is an advantage over the imaging approach. Various laparoscopic techniques with dedicated instrumentation for isolating the cyst provide additional protection during the procedure [45,46].

Additionally, laparoscopy represents the only minimally invasive option in settings where the PAIR approach is not available.

Regarding open surgery, it remains an option for locations inaccessible through the PAIR or laparoscopic approach in cases of hydatid complications (rupture into the abdominal cavity, biliary complications, septic or anaphylactic complications) or intraoperative/intraprocedural complications during laparoscopy or PAIR [47,48,49,50].

Mesenteric, peritoneal, epiploic, or round ligament locations are rarely amenable to laparoscopic approaches and are almost exclusively addressed through open surgery, especially when they are unique and accessible.

Open surgery is the preferred surgical approach for intra-abdominal extrahepatic locations. This preference is based on the difficulty of inactivating the cystic content and the risk of intraoperative contamination.

Another consideration is the challenge of imaging exposure in PAIR due to the overlapping segments of the digestive tract, hindering clear visualization of the cystic formation.

When surgically addressing a formation with such localization, the goal is the complete excision of the cyst, sometimes without attempting to inactivate it, a maneuver that, in the case of laparoscopy, may predispose to spillage.

Laparoscopic surgery for hydatid cysts, both with hepatic and intra-abdominal locations, should be considered even in cases of multiple or challenging positions. The results are significantly superior to open surgery, and the risk of spillage can be mitigated through accurate imaging, good pericystic isolation, and careful dissection.

Additional imaging data beyond standard examinations that only involve CT or ultrasound, using artificial intelligence models, represent a milestone in terms of diagnosis and localization.

These methods leverage images obtained through ultrasound and tomographic examinations, applying artificial intelligence techniques and 3D or 4D model reconstruction, providing high-fidelity data regarding the type of cystic lesion, the involved parasite, the relationship with hepatic parenchyma, the vital structures of the liver, and the relationship with neighboring organs [14,51].

These techniques find applicability in distinguishing a hydatid cyst from other hepatic cystic conditions, especially biliary mucinous cystic neoplasm, where the surgical approach differs from the oncological perspective. Despite being a parasitic disease, hydatid cysts can behave like a neoplasia, with invasive, debilitating characteristics and a tendency to complicate the affected organs. Through distant dissemination from the primary lesion, it can exhibit metastatic characteristics [52,53].

## 5. Case Report

We present the case of a 45-year-old female patient with a BMI of 26.08, radiologically diagnosed with hepatic, mesenteric, and peritoneal hydatid cysts. After initiating therapy with albendazole 400 mg/day, the patient sought the surgical department for further evaluation.

A CT examination revealed multiple cystic images in the hepatic segments V, VI, and VII, with subhepatic extension and involvement at the fissures of the venous ligament. Cystic images were also observed in the anterior abdominal wall and submesocolic (Figure 5, Figure 6, Figure 7, Figure 8 and Figure 9). Blood tests showed no significant abnormalities, and ELISA determination of IgG class antibodies to *Echinococcus granulosus* was positive at 1.66 IV (with values considered negative at ≤0.9 IV and positive at ≥1.1 IV).

American Society of Anesthesia Risk Classification II was assigned to our patient, who was proposed for a complex surgical procedure that required general anesthesia with endotracheal intubation. The induction phase was uneventful and involved the sequential administration of sufentanil, propofol, and succinylcholine. The maintenance phase was also uneventful, and a combination of sufentanil, rocuronium, and sevoflurane was used to ensure optimal anesthesia throughout the procedure.

The surgical intervention was performed with the patient in a supine position, with the surgical team placed in an American position, alternating between the Fowler and Trendelenburg patient positions based on intraoperative needs. Carbon dioxide pneumoperitoneum was achieved through a supraumbilical approach using a Veress needle while maintaining the intraabdominal pressure at 12.13 mmHg. Laparoscopic exploration (using a 30° laparoscope) was carried out through trocar placement, including a 10 mm trocar supraumbilically and epigastrically and 5 mm trocars in the left and right flanks and the left iliac fossa.

The imaging highlights hepatic cysts in segments V, VI, and VII, a large cystic formation in the round ligament, a cyst in the parietal peritoneum, two cysts in the greater omentum, and two cysts located in the submesocolic enteral mesentery.

The cyst located in segment V shows progression into segment IV B, with involvement of the hepatic parenchyma, the gallbladder being intimately adherent to the cystic wall (Figure 6).

Using LigaSure, laparoscopic excision of the cyst at the round ligament level is performed, followed by the excision of the peritoneal cysts, including those in the greater omentum and mesentery. The excision is carried out with the intact capsule, and the formations are placed in a closed endobag and left in the pelvis.

The surgical hepatic area of interest is isolated with alcohol-soaked gauze. Laparoscopic cholecystectomy is performed, and the cyst at this level is punctured, aspirated, inactivated with a hypertonic solution, and then reaspirated with the evacuation of cystic contents, daughter vesicles, and the proligera membrane. The cysts in segments VI and VII are approached in the same manner, with partial pericystectomy for two and total cystectomy for two others. A pedunculated formation on the anterior surface of the uterus is identified and excised (Figure 10, Figure 11, Figure 12 and Figure 13).

Using an endobag, the cysts are extracted from the abdominal cavity through a minimal suprapubic incision in the Kustner variant. Drainage of the hepatic cyst cavities and drainage at the level of the Douglas pouch is performed.

The postoperative course is uneventful, and the patient is discharged five days postoperatively, continuing antiparasitic treatment for another month. The histopathological result highlights hydatid cysts, uterine leiomyomatous fibroids without atypia, and a gallbladder with cholesterolosis changes.

Imaging follow-up at 3 and 9 months does not reveal signs of recurrence.

## 6. Conclusions

The concurrent localizations of hydatid disease in the liver, round ligament, peritoneum, mesentery, and omentum are rare and pose a surgical management challenge. The inconvenience of multiple localization and the risk of contamination usually lead to open surgery in these cases. The presented case is unique due to the laparoscopic approach to the multiple localizations of hydatid disease, advocating for a minimally invasive first-line approach even in these particular localizations.

## Figures and Tables

**Figure 1 jpm-14-00205-f001:**
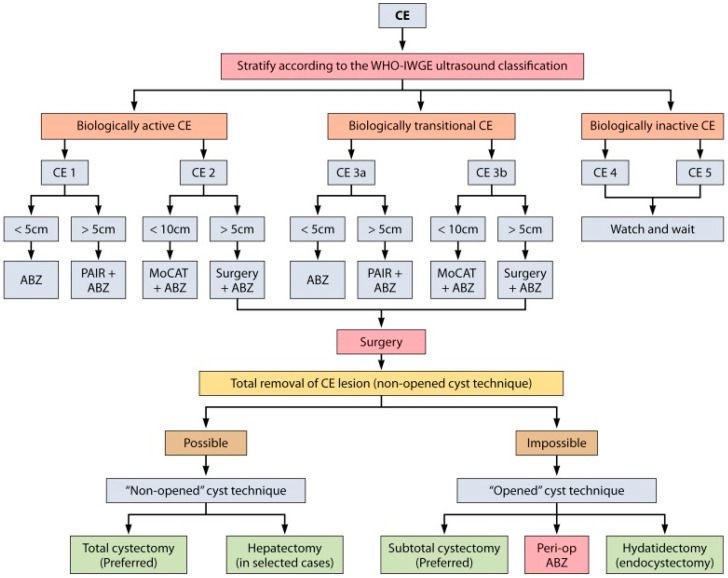
Treatment algorithm in hydatid cyst based on the WHO-IWGE international classification of ultrasound images in cystic echinococcosis for application in clinical and field epidemiological settings (CE = cystic echinococcosis; ABZ = albendazole; MoCAT = Modified Catheterisation Technique; PAIR = Puncture, Aspiration, Instillation, Reaspiration) [17].

**Figure 2 jpm-14-00205-f002:**
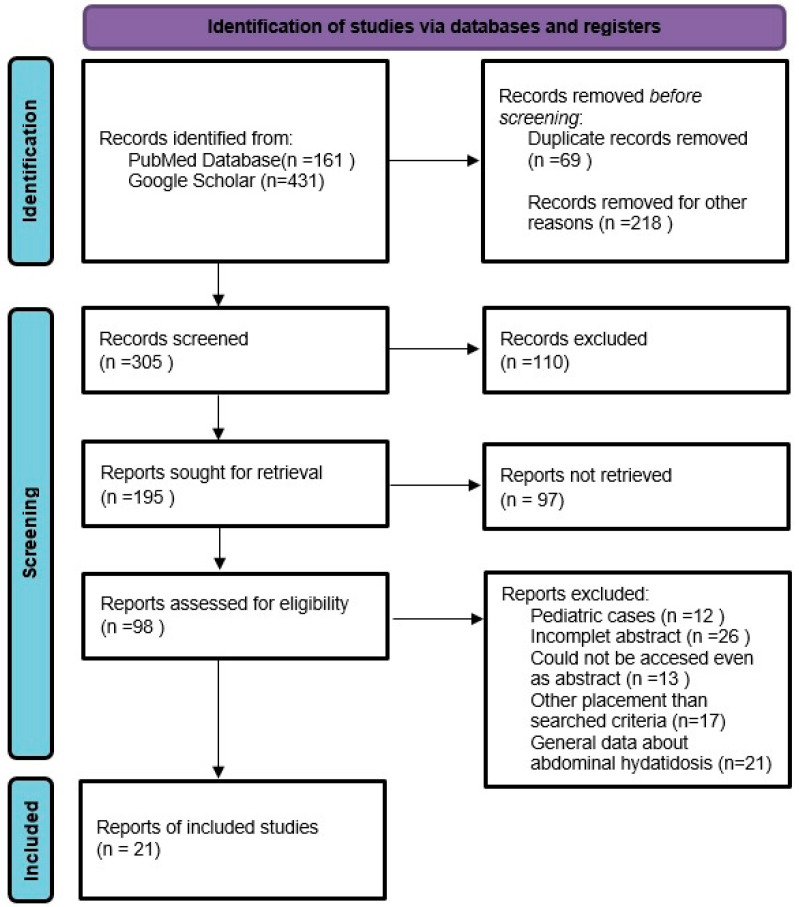
PRISMA flow diagram for this study.

**Figure 3 jpm-14-00205-f003:**
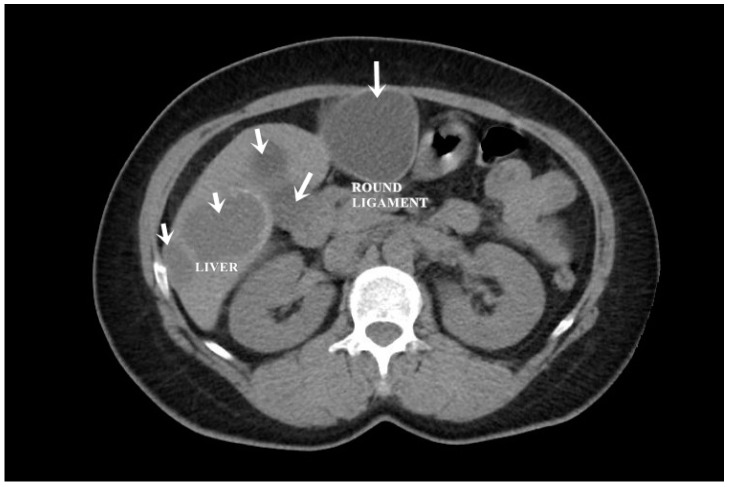
Liver and round ligament hydatid cysts (indicated by arrows).

**Figure 4 jpm-14-00205-f004:**
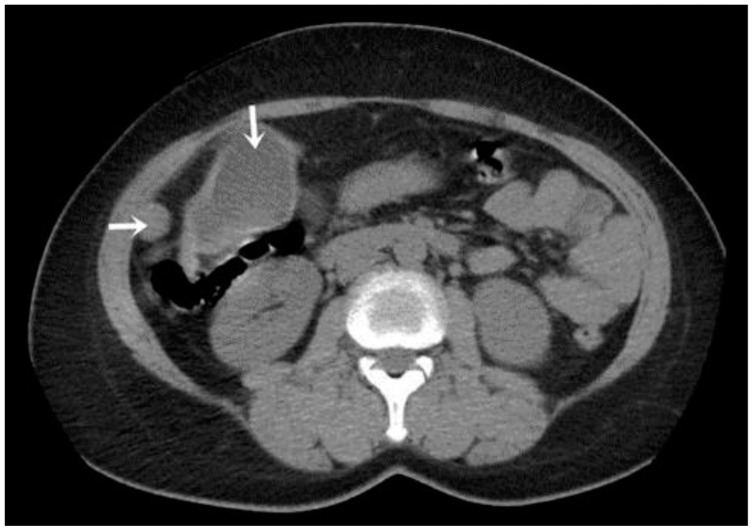
Peritoneal and segment V cyst (indicated by arrows).

**Figure 5 jpm-14-00205-f005:**
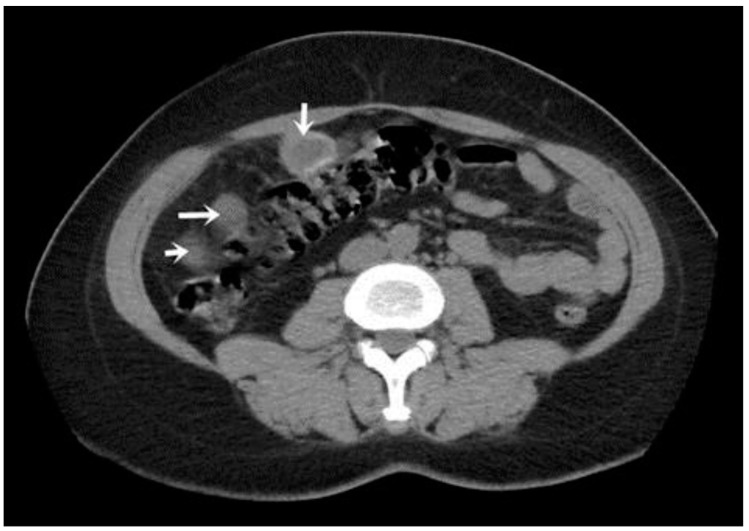
Omental hydatid cysts (indicated by arrows).

**Figure 6 jpm-14-00205-f006:**
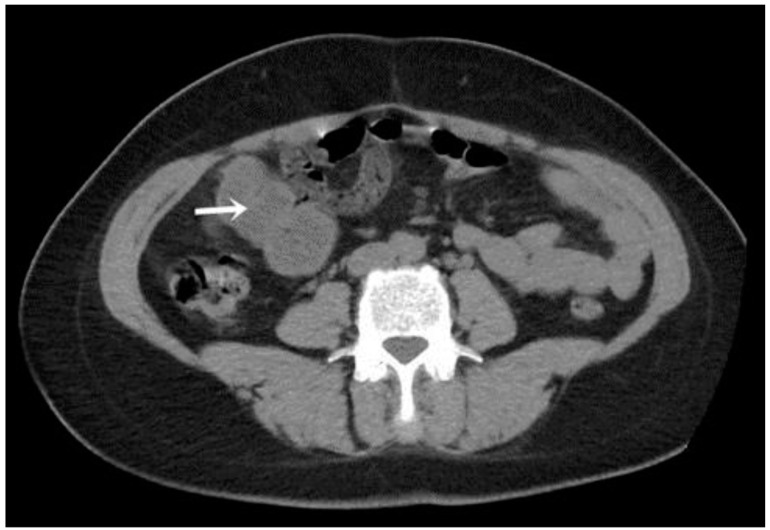
Mesenteric hydatid cyst (indicated by arrow).

**Figure 7 jpm-14-00205-f007:**
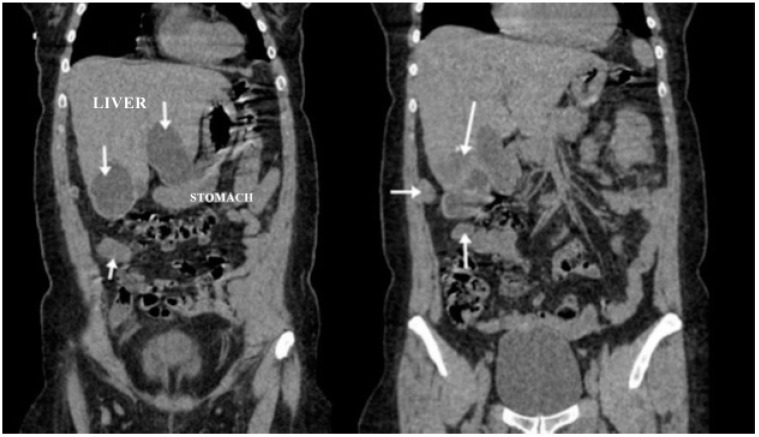
Coronal section (liver, round ligament, mesenteric, omental, and peritoneal hydatid cysts indicated by arrows).

**Figure 8 jpm-14-00205-f008:**
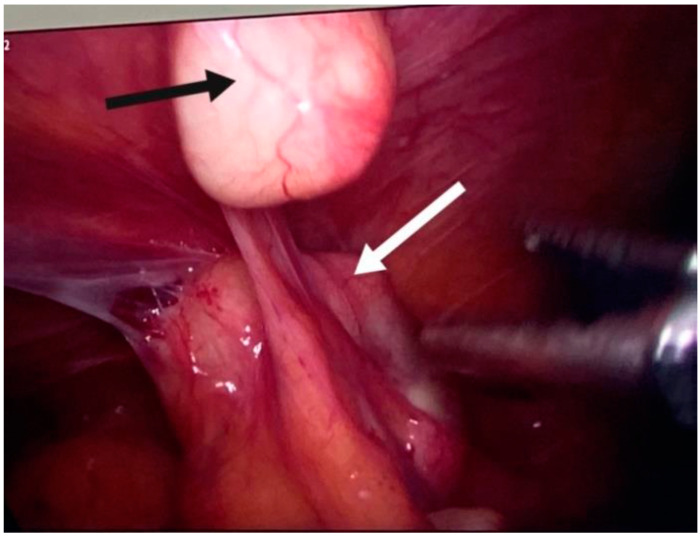
Peritoneal (black arrow) and round ligament cysts (white arrow).

**Figure 9 jpm-14-00205-f009:**
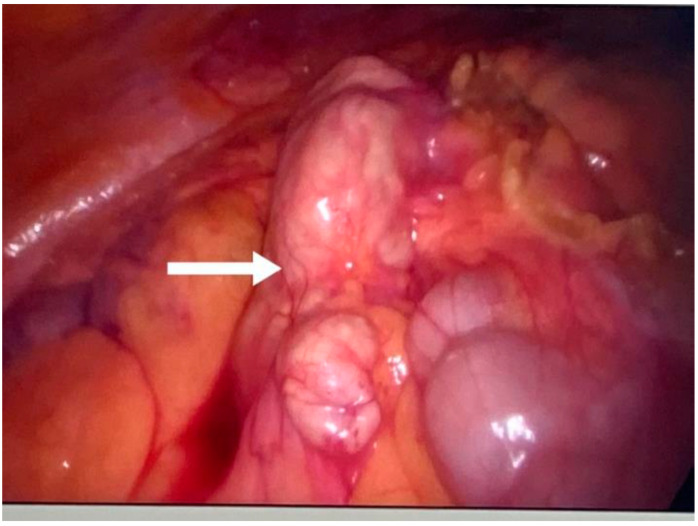
Omental hydatid cyst (white arrow).

**Figure 10 jpm-14-00205-f010:**
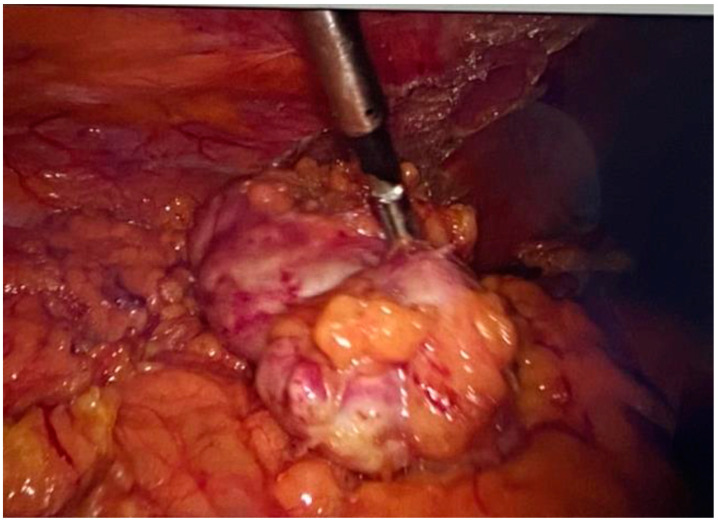
Resected omental specimen.

**Figure 11 jpm-14-00205-f011:**
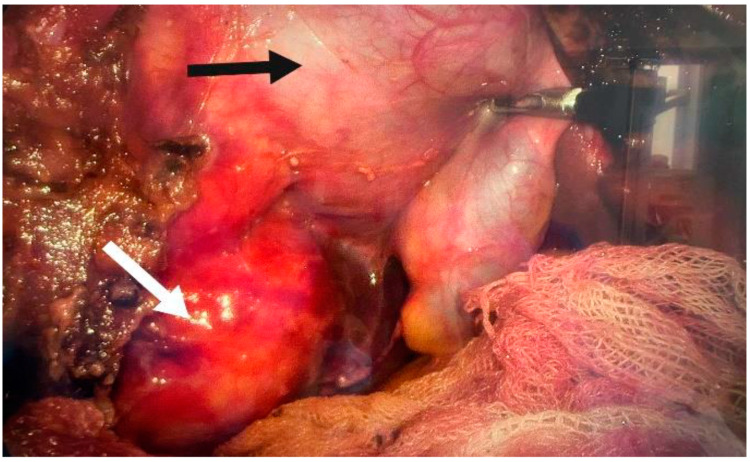
Hepatic hydatid cyst (white arrow) and gallbladder (black arrow).

**Figure 12 jpm-14-00205-f012:**
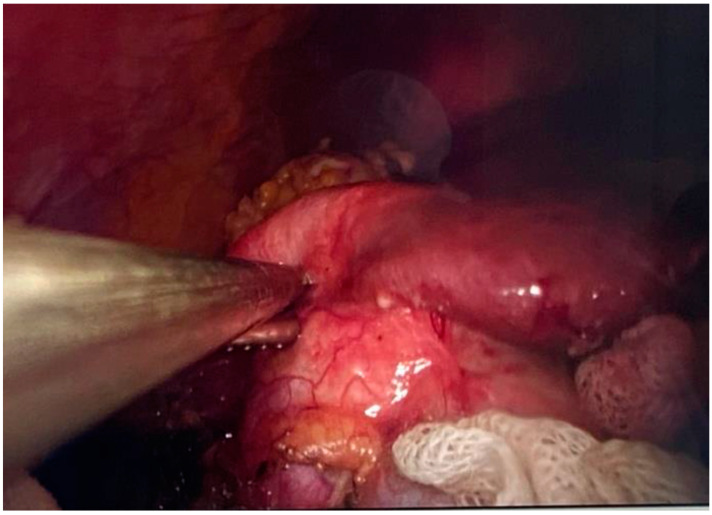
Hepatic hydatid cyst aspiration.

**Figure 13 jpm-14-00205-f013:**
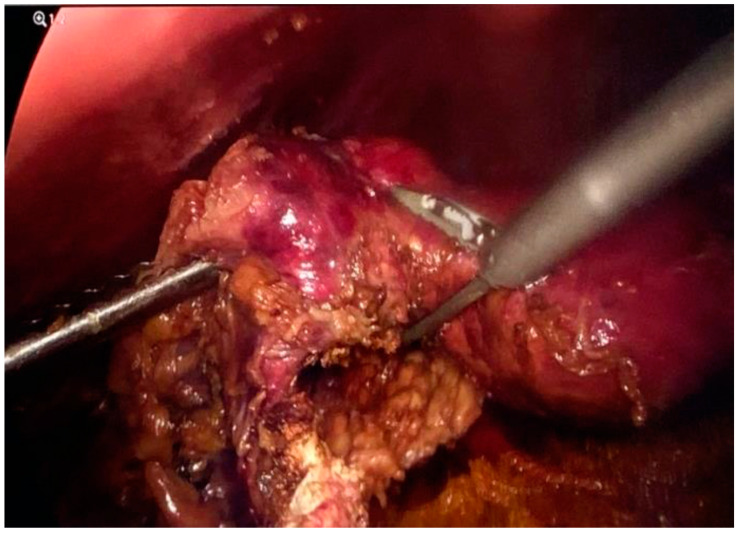
Hepatic hydatid cyst resection.

**Table 1 jpm-14-00205-t001:** Round ligament, omental, and mesenteric hydatid cysts—literature review.

Authors	Age (Years)	Gender	Symptoms	Imaging	Preoperative Treatment	Hydatid Cyst Placement	Inactivation Solution	Surgical Management	Outcome
Kusaslan R. et al. [19]	19	M	Acute abdominal pain	Ultrasound	-	Mesentery	10% povidone-iodine	Laparotomy, drainage	Favorable
Sachar S. et al. [20]	40	F	Abdominal pain	Ultrasound	-	Mesentery	Hypertonic saline	Laparotomy, excision	Favorable
Kushwaha J. K. et al. [7]	24	M	Abdominal mass, abdominal pain	Ultrasound	Albendazole 1 month	Mesentery	10% povidone-iodine	Laparotomy, excision	Favorable
Abdelmaksoud M. M. et al. [21]	91	M	Abdominal pain	CT scan	Albendazole 3 weeks	Mesenterynand spleen	Hypertonic saline	Laparotomy, splenectomy, partial excision	Favorable
Paramythiotis D. et al. [22]	39	M	Acute abdominal pain	Ultrasound/CT scan	-	Mesentery and liver	Hypertonic saline	Laparotomy, drainage, and excision	Favorable
Ranjan R. et al. [23]	39	F	Abdominal pain	Ultrasound/CT scan		Mesentery	-	Laparotomy, excision	Favorable
Adelyar M. A. et al. [24]	40	F	Abdominal pain	-	Albendazole 1 month	Mesentery	-	Laparotomy, excision	Favorable
Fasihi Karami M. et al. [25]	51	M	Abdominal mass, abdominal pain	Ultrasound/CT scan	-	Omentum and liver	-	Laparotomy, drainage	Favorable
Jangjoo A. et al. [26]	33	F	Abdominal pain	CT scan	-	Omentum	-	Laparoscopy, excision	Favorable
Sable S. et al. [27]	52	M	Abdominal pain	Ultrasound/CT scan	Albendazole	Omentum	-	Laparotomy, excision	Favorable
Leitão P. et al. [27]	49	M	Abdominal pain	CT scan MRI	3 months of Albendazole	Omentum and liver	-	Laparotomy, hepatectomy, pericystectomy	Favorable
Aghaei A. et al. [28]	82	F	Abdominal mass	CT scan	-	Omentum	10% povidone-iodine	Laparotomy, partial excision	Favorable
Akinci M. et al. [29]	24	M	Abdominal mass	Ultrasound	-	Round ligament	-	Laparotomy, excision	Favorable
Velioglu M. et al. [30]	21	M	Abdominal mass	CT scan	Albendazole	Mesentery	-	Laparotomy, excision	Favorable
Talatnoor S. et al. [31]	46	F	Abdominal mass	Ultrasound/CT scan	-	Mesentery	-	Laparotomy, excision	Favorable
Jatal S. N. et al. [32]	30	M	Abdominal mass	Ultrasound	-	Mesentery	10% povidone-iodine	Laparotomy, partial excision	Favorable
Duzkoylu Y. et al. [33]	62	M	Abdominal pain abdominal mass	Ultrasound/CT scan	-	Mesentery	-	Laparotomy, excision	Favorable
Mittal S. et al. [34]	35	F	Abdominal pain	Ultrasound	-	Mesentery	-	Laparotomy, excision	Favorable
Ghafouri A. et al. [35]	21	F	Abdominal pain	Ultrasound/CT scan	-	Omentum	-	Laparotomy, excision	Favorable
Ertugrul I. et al. [36]	49	M	Abdominal mass	CT scan	-	Omentum	-	Laparoscopy, excision	Favorable
Busić Z. et al. [37]	42	M	Abdominal pain	CT scan	Albendazole	Liver, omentum, peritoneum	-	Laparoscopy, excision	Favorable

**Table 2 jpm-14-00205-t002:** Relation between hydatid cyst placement and symptoms (variables with statistical significance were marked in tables with *).

		Acute Abdominal Pain		Abdominal Pain		Abdominal Mass, Abdominal pain		Abdominal Mass		Total	
		n	%	n	%	n	%	n	%	n	%
Hydatid Cyst Placement	Mesentery	2	9.52% *	4	19.05%	2	9.52% *	3	14.29%	11	52.38%
	Omentum and liver	0	0%	1	4.76%	1	4.76%	0	0%	2	9.52%
	Omentum	0	0%	3	14.29%	0	0%	2	9.52%	5	23.81%
	Round ligament	0	0%	0	0%	0	0%	1	4.76%	1	4.76%
	Liver, omentum, peritoneum	0	0%	1	4.76%	0	0%	0	0%	1	4.76%
	Mesentery and spleen	0	0%	1	4.76%	0	0%	0	0%	1	4.76%
	Total	2	9.52%	10	47.62%	3	14.29%	2	28.57%	21	100%

**Table 3 jpm-14-00205-t003:** Relation between hydatid cyst placement and surgical technique (variables with statistical significance were marked in tables with *).

Surgical Management									
		Laparotomy, Drainage	Laparotomy, Excision	Laparotomy, Splenectomy, Partial Excision	Laparotomy, Drainage, and Excision	Laparoscopy, Excision	Laparotomy, Hepatectomy, Pericystectomy	Laparotomy, Partial Excision	Total
Hydatid cyst placement	Mesentery	4.76% *	38.1%	0%	0%	0%	0%	4.76% *	47.62%
	Mesentery and liver	0%	0%	0%	4.76%	0%	9%	0%	4.76%
	Omentum and liver	4.76%	0%	0%	0%	0%	4.76%	0%	9.52%
	Omentum	0%	9.52%	0%	0%	9.52%	0%	4.76%	23.81%
	Round ligament	0%	4.76%	0%	0%	0%	0%	0%	4.76%
	Liver, omentum, peritoneum	0%	0%	0%	0%	4.76%	0%	0%	4.76%
	Mesentery and spleen	0%	0%	4.76%	0%	0%	0%	0%	4.76%
	Total	9.52%	52.38%	4.76%	4.76%	14.29%	4.76%	9.52%	100%

## Data Availability

Not applicable.

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
