# Peer review of "Laparoscopic Management of Multiple Liver, Omental, Mesenteric, Peritoneal, and Round Ligament Hydatid Cysts—A Rare Report of a Case and a Systematic Literature Review"

_jpm, 2024, doi:10.3390/jpm14020205_

Round 1

Reviewer 1 Report

Comments and Suggestions for Authors

Laparoscopic management of multiple liver, omental, mesenteric, peritoneal and round ligament hydatid cysts - rare report of a case and systematic literature review.

The presented case is unique due to the laparoscopic approach to the multiple localizations of hydatid disease, advocating for a minimally invasive first-line approach even in these particular localizations.

Excellent report.

Minor mistakes

l.31 in intro, pl use italics for species names.

l.319, l. 339, pl keep space between numbers and units

Figures

Please, enrich the figures with legends and better description. People with Anatomy founded knowledge are not able to follow.

References

Pl edit all of your refs and adapt to the journals style- there are irregularities and not homogeneity in the citation style.

Comments on the Quality of English Language

noted : minor improvements.

Author Response

First and foremost, we would like to express our gratitude for the effort and time you have dedicated to reviewing our article. We are pleased that you have found our case to be unique and worthy of publication.

Minor mistakes

l.31 in intro, pl use italics for species names.

Thank you for the observation. We have formatted the text by using italics within the suggested line, according to the guidelines.

l.319, l. 339, pl keep space between numbers and units

Thank you. We have made the changes according to your suggestion.

Figures

Please, enrich the figures with legends and better description. People with Anatomy founded knowledge are not able to follow.

We appreciate the suggestion you provided. We have enriched the description of the figures in the text by attempting to present the anatomical peculiarities of the described case in a manner that is as accessible as possible for readers.

References

Pl edit all of your refs and adapt to the journals style- there are irregularities and not homogeneity in the citation style.

Thank you for bringing this to our attention. We acknowledge the need to ensure consistency and adherence to the journal's citation style throughout our references. We promptly addressed the irregularities and ensure that all citations are formatted correctly to maintain the desired level of homogeneity.

Reviewer 2 Report

Comments and Suggestions for Authors

The work of Alin Mihetiu and collaborators is well presented and designed.

The authors carried out a good review of the literature, presented the clinical case excellently, and carried out the corresponding discussion.

One of the few jobs that I review that are well designed and carried out correctly.

congratulations

Author Response

Above all, we wish to convey our heartfelt appreciation for the dedication and time you have generously invested in reviewing our article. It brings us great joy to learn of your recognition of the distinctive nature and publication-worthy quality of our case!

Reviewer 3 Report

Comments and Suggestions for Authors

This study has merit and will be interesting for readers who are medical doctors or biologists interested in hydatid cysts. However, the written style used in the manuscript can be improved. There are many small paragraphs in all sections except the abstract. So, I suggest the authors must connect all these gaps together to make the story smoother and easier to read.

There are points that need improvement to make the manuscript better:

Abstract: Needs improvement by pointing out that you conducted a systematic review of laparoscopic management of hydatid cysts and the reasons why you had to do that.

Introduction:

Italicize all the scientific names (line 31: Echinococcus granulosus).

Line 34: “The tapeworm lives as a definitive host…” Correct this sentence.

Line 36: Change "Echinococcus Granulosus and Multilocularis" to "E. granulosus and E. multilocularis."

Lines 39-41: Need a reference to support this sentence.

Lines 46-47: Need a reference to support this sentence.

Lines 72-74: Add references for this sentence.

Lines 84-86: Need a reference to support this sentence or connect this paragraph to the next one.

The authors should address more about the gaps that need to be added to the scientific community and why a systematic review is needed to clarify some points regarding hydatid cysts.

The objectives of the study should be written in the introduction section.

It is difficult for readers to understand all abbreviations in Figure 1.

Materials and Methods:

Lines 72-74: Change “Scholar” to “Google Scholar” throughout the manuscript.

The paragraph at lines 123-125 should come last after describing screening the databases.

In Table 1, check the references for Abdelmaksoud MM et al (21) and Paramythiotis D et al (21).

Results:

Line 167: Specify whether the age is in years or months.

The odds ratio (OR), p-value, etc., are described in the text. The authors should include these in their Tables (in the manuscript, these are Figure 3 and Figure 4). Include some symbols to indicate statistical significance.

Some interesting points in the results can be mentioned in the abstract.

In Figures 5-13, there are many arrows without description. Please provide more details about what those arrows represent.

Discussion:

The numerous small paragraphs make me feel like I am reading a textbook.

Comments on the Quality of English Language

Can be improved.

Author Response

Primarily, we extend our deepest gratitude for the invaluable time and effort you have dedicated to reviewing our article.

There are many small paragraphs in all sections except the abstract. So, I suggest the authors must connect all these gaps together to make the story smoother and easier to read.

   Thank you for your insightful suggestion. We acknowledge the importance of enhancing the coherence and readability of our manuscript. We will endeavor to connect the smaller paragraphs in all sections to create a smoother and more engaging narrative for our readers. Your feedback is greatly appreciated.

There are points that need improvement to make the manuscript better:

Abstract: Needs improvement by pointing out that you conducted a systematic review of laparoscopic management of hydatid cysts and the reasons why you had to do that.

   We have taken into consideration your suggestions to enhance the clarity and depth of our manuscript. Specifically, we revised the abstract to clearly indicate that our study involved a systematic review of laparoscopic management techniques for hydatid cysts. Additionally, we provided a thorough explanation of the reasons behind conducting this review, emphasizing the importance of addressing existing gaps in the literature. We have denoted the modifications in yellow for clarity and ease of identification.

Introduction:

Italicize all the scientific names (line 31: Echinococcus granulosus).

   Thank you for the observation. We have formatted the text by using italics within the suggested line.

Line 34: “The tapeworm lives as a definitive host…” Correct this sentence.

   We have rectified the indicated sentence.

Line 36: Change "Echinococcus Granulosus and Multilocularis" to "E. granulosus and E. multilocularis."

   Thank you. We have made the changes according to your suggestion.

Lines 39-41: Need a reference to support this sentence.

Lines 46-47: Need a reference to support this sentence.

Lines 72-74: Add references for this sentence.

Lines 84-86: Need a reference to support this sentence or connect this paragraph to the next one.

   We have added a reference for each paragraph as specified within the review. Thank you for your guidance.

The authors should address more about the gaps that need to be added to the scientific community and why a systematic review is needed to clarify some points regarding hydatid cysts.

   Thank you for your suggestion! We have added a paragraph highlighting the contribution our article brings to the existing specialized literature and the new frontier it promotes, namely the minimally invasive approach in hydatid cyst surgery with numerous secondary disseminations.

The objectives of the study should be written in the introduction section.

   We have added a new paragraph in the introduction section, aimed at specifically highlighting the objectives that the study aims to achieve.

It is difficult for readers to understand all abbreviations in Figure 1.

   Thank you. We have provided explanations for the abbreviations in Figure 1, adhering to the provided instructions.

Materials and Methods:

Lines 72-74: Change “Scholar” to “Google Scholar” throughout the manuscript.

   Thank you. We have made the changes according to your suggestion.

The paragraph at lines 123-125 should come last after describing screening the databases.

   We have repositioned the indicated paragraph within the text to align with the provided instructions. Thank you.

In Table 1, check the references for Abdelmaksoud MM et al (21) and Paramythiotis D et al (21).

   Thank you for addressing the identified error. We have subsequently rectified the indicated mistake.

Results:

Line 167: Specify whether the age is in years or months.

   We have supplemented the table's second column with details regarding the expression of age in years.

The odds ratio (OR), p-value, etc., are described in the text. The authors should include these in their Tables (in the manuscript, these are Figure 3 and Figure 4). Include some symbols to indicate statistical significance.

   The odds ratio (OR), p-value, etc., have been integrated into Table 2 as per your suggestion. Symbols indicating statistical significance - asterisks (*) have also been included for enhanced clarity.

Some interesting points in the results can be mentioned in the abstract.

   Several noteworthy findings from the results have been incorporated into the abstract for improved comprehensiveness and interest.

In Figures 5-13, there are many arrows without description. Please provide more details about what those arrows represent.

   Thank you! The arrows lacking descriptions have been supplemented with additional details clarifying their representations.

Discussion:

The numerous small paragraphs make me feel like I am reading a textbook.

   We have implemented changes to facilitate a more accessible reading experience. By refining the structure and content, we aimed to enhance readability and ensure clarity for readers. We extend our sincere gratitude for your valuable feedback and constructive suggestions.

Round 2

Reviewer 3 Report

Comments and Suggestions for Authors

Overall, I am satisfied with this revised version of the manuscript. However, there are some minor points to make it better.

Line 45: Change “Echinococcus Granulosus and Multilocularis” to "Echinococcus granulosus and Echinococcus multilocularis". The first letter of the species' names must not be capitalized.

Figure 3: Add “(indicated by arrows)” if those arrows indicate the liver and round ligament hydatid cysts.

Figure 4: Add “(indicated by arrows)” if those arrows indicate the peritoneal and segment V cyst.

Figure 8: Add “(white arrows)” after "round ligament cysts" if the arrow indicates the round ligament cyst.

Figure 9: Add “(white arrow)” after "Omental hydatid cyst" if the arrow indicates the omental hydatid cyst.
